# Developing an Attention-Based Deep Learning Framework for Obstructive Sleep Apnea Detection Using Single-Channel Oximetry Signal

Malar Paavai Muthukumaran[1†], Micky C. Nnamdi[1†], J. Ben Tamo[1], Chad Purnell[2], May D. Wang[1]

[1]*Georgia Institute of Technology*, Atlanta, USA
[2]*Shriner's Hospitals for Children*, Chicago, USA
{mpm9, mnnamdi3, jtamo3, maywang}@gatech.edu, cpurnell@shrinenet.org

*Abstract*—**Obstructive Sleep Apnea (OSA) is a common yet underdiagnosed condition typically assessed using polysomnography, a resource-intensive procedure. Oximetry ($SpO_2$) offers a non-invasive, low-cost alternative for large-scale OSA screening. This study proposes an interpretable deep learning framework for estimating the Apnea-Hypopnea Index (AHI) and classifying OSA severity using only single-channel $SpO_2$ signals. The model integrates convolution, bidirectional long short-term memory, graph attention networks, and multi-head attention to capture both local and global temporal patterns. Model predictions are interpreted using class-specific attention heatmaps and residual analysis. Evaluated on two large datasets (SHHS and CFS), the model achieved strong performance, with $R^2$ values ranging from 0.868 to 0.941 and outperformed baseline models across $R^2$, F1 score, sensitivity, and precision. Confusion matrices showed high classification accuracy for No Apnea and Severe cases, while scatter plot and Bland–Altman analyses confirmed low bias and stable predictions. These results demonstrate that $SpO_2$-based models can provide accurate and scalable AHI estimation, with attention-based visualizations enhancing interpretability and supporting clinical screening without the need for full PSG.**

*Index Terms*—**Obstructive sleep apnea, graph attention, multi-head attention, clinical decision system, oximetry signal**

## I. INTRODUCTION

Obstructive sleep apnea (OSA) affects an estimated one billion individuals worldwide between the ages of 30 and 65, underscoring its significant negative impact on public health [1]. It represents the most common form of sleep-disordered breathing [1]. If left untreated, OSA is associated with a range of adverse health outcomes, including cardiovascular disease, metabolic dysfunction, depression, and cognitive impairment [2]. Despite these risks, many cases remain undiagnosed and untreated [2].

The apnea-hypopnea index (AHI) is commonly used to quantify OSA by representing the number of apnea and hypopnea events per hour of sleep. Based on the calculated AHI, clinicians classify OSA severity as normal if values are below 5, mild if values are $\geq 5$ and $< 15$, moderate if values are $\geq 15$ and $< 30$, and severe if values are $\geq 30$ [3]. AHI is typically derived from polysomnography (PSG), the gold standard for OSA diagnosis. PSG requires multiple sensors, a lengthy setup, and overnight monitoring, making it resource-intensive, time-consuming, and costly [4]. To address these limitations, alternative methods such as portable monitoring devices and home sleep apnea tests have been developed. These range from multi-channel Type II/III systems to simplified Type IV monitors with one or two physiological signals [5]–

[7]. Among these, oximetry, which tracks oxygen saturation ($SpO_2$), is a particularly informative and accessible biomarker for OSA [8].

Several studies have introduced oximetry-based biomarkers that capture distinct patterns within the signal, such as approximate entropy, detrended fluctuation analysis, and desaturation-based indicators [9]. Furthermore, previous research has demonstrated that simple oximetry-based screening has the ability to classify OSA severity [9]. $SpO_2$ can be conveniently measured at home using finger sensors or wearable devices, making it an accessible and affordable option for OSA detection.

With the growth of large open-source sleep datasets, such as those from the National Sleep Research Resource (NSRR), research has increasingly turned to data-driven methods for OSA detection. Traditional approaches rely on manually engineered features, such as the oxygen desaturation index (ODI), for classification. For example, a study based on manually engineered features for empirical model decomposition (EMD) of $SpO_2$ signals achieved a sensitivity and specificity of 0.838 and 0.855, respectively, demonstrating the potential of oximetry-based screening for OSA detection without the need for complex instrumentation [10].

Recently, deep learning algorithms have gained significant momentum due to their ability to automatically extract complex patterns from oximetry signals that may be overlooked by manually engineered features, achieving better performance compared to traditional classifiers [11]. Convolutional neural networks (CNNs), recurrent neural networks (RNNs), long short-term memory networks (LSTMs), and, more recently, transformers and attention-based models have been applied to $SpO_2$ signal for apnea event detection, AHI estimation, and OSA severity classification [12], [13]. Hybrid models such as OxiNet, which combines CNNs and convolutional recurrent neural networks (CRNNs), have outperformed benchmark models [9]. These advances highlight the potential of deep learning to capture both local features and long-term temporal patterns in oximetry data. However, the lack of interpretability and limited insight into how predictions are made raises concerns in clinical settings, where understanding the basis of a model's decision is critical for use and trust. Therefore, recent research has started addressing this by using attention mechanisms and explainability techniques or visualizations to highlight which parts of the signal influence the model's decision [14], [15]. However, relatively few models focus on oximetry-only inputs and systematically analyze how attention

---

[†] The first two authors contributed equally to this work.

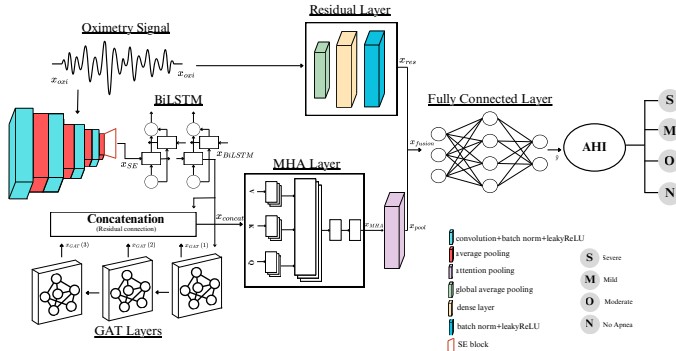

Fig. 1. The overall proposed network architecture for classifying the presence and severity of OSA. Conv1D layers and the squeeze-and-excitation (SE) block are employed for feature extraction, while bidirectional LSTM layers capture temporal dependencies. Graph attention network (GAT) layers, a multi-head attention (MHA) layer, and attention pooling form the attention mechanism. A residual layer incorporates global contextual information. The remaining components are designed for AHI prediction and severity classification.

behavior differs across AHI-defined severity levels.

In this study, we propose an interpretable deep learning model (Figure 1) that combines convolutional layers, graph attention network (GAT), and multi-head attention (MHA) to estimate the AHI directly from $SpO_2$ signals. Beyond achieving strong predictive performance, our work emphasizes interpretability and clinical relevance. The key contributions of this work are as follows:

- **End-to-end architecture for $SpO_2$-based AHI estimation:** We introduce a novel deep learning framework that integrates convolution, GAT, and MHA to accurately predict AHI from single-channel pulse oximetry signals.
- **Interpretable attention-based modeling:** By incorporating both GAT and MHA layers, our model highlights relevant temporal regions through attention visualization, offering clinically meaningful insights into desaturation patterns associated with different OSA severity levels.
- **Severity-aware evaluation and single-modality inference:** We demonstrate that our model can effectively distinguish OSA severity levels using only $SpO_2$ data, achieving accurate and explainable performance without reliance on additional physiological signals.

## II. RELATED WORKS

### A. Deep Learning for $SpO_2$-based OSA detection

Deep learning models have shown great promise in automating the detection of OSA from biophysiological signals, particularly $SpO_2$. CNNs are especially effective at learning desaturation patterns directly from raw $SpO_2$ signals due to their ability to extract local temporal features. For instance, Albuhayri et al. [13] developed an 8-layer CNN to classify apnea events with promising results. More recent advancements include the use of ResNet-style CNNs that incorporate multi-scale and residual learning techniques. These models help mitigate issues such as the vanishing gradient problem, thereby enhancing training stability and performance [16]. While CNNs are computationally efficient and well-suited for short-term pattern recognition, they often fall short in modeling long-term temporal dependencies, an important aspect for accurate AHI estimation.

To overcome this limitation, hybrid architectures that com-

bine CNNs with RNNs, such as LSTM and gated recurrent unit (GRU) models, have been explored. These hybrid models leverage convolutional layers for localized feature extraction and recurrent layers for sequence modeling over time. A notable example is OxiNet, proposed by Lévy et al., which uses a dual-branch architecture, one purely CNN-based and the other combining CNN with RNN, to predict AHI from overnight $SpO_2$ data, outperforming traditional indices such as the ODI [9]. Other CNN-LSTM approaches have similarly demonstrated improved performance by capturing the temporal evolution of desaturation patterns.

### B. Attention Mechanisms and Explainability in Deep Learning Models for OSA

Recent advancements in deep learning for OSA detection have focused on attention-based models, particularly transformers, to address the limitations of RNNs. Transformers leverage MHA mechanisms to model long-range dependencies in $SpO_2$ signals, making them well-suited for capturing the periodic yet irregular nature of OSA events. For example, Almarshad et al. [17] introduced a pure transformer architecture with learnable positional encoding for OSA detection outperforming conventional CNN-based models on the Obstructive Sleep Apnea Stroke Unit Dataset (OSASUD). The ability of attention heads to highlight both localized desaturations and broader temporal trends enhances the detection of subtle or irregular apnea events. In addition to pure transformer models, attention mechanisms have been incorporated into hybrid CNN-RNN architectures. One such model, the CNN-BiGRU with attention, demonstrated improved classification of pediatric OSA severity by focusing on salient temporal features [16].

Attention has also been explored in multimodal settings, where $SpO_2$ signals are combined with other physiological modalities such as ECG, airflow, or audio. Co-attention mechanisms, for instance, align desaturation patterns with heart rate variability to enhance diagnostic accuracy [18]. GAT has also emerged as a promising direction. For example, TF-GAT extracts information from both the frequency domain and time series, and it uses these features for graph data [19]. It combines GAT and MHA to capture dependencies between multimodal time series and time-frequency signal relationships [19].

Despite these advances, few studies have focused on incorporating attention mechanisms like GAT into single-channel $SpO_2$-only models with an emphasis on interpretability. This highlights a significant gap in the literature and motivates our work, which aims to develop accurate and explainable OSA detection models using single-channel pulse oximetry alone.

## III. METHODOLOGY

### A. Problem Formulation

Given $X = \{x_1, x_2, ..., x_T\} \in \mathbb{R}^T$ represent a univariate time series corresponding to the overnight oximetry ($SpO_2$) signal, sampled at 1 Hz over $T$ seconds. The objective is to learn a function $f_\theta : \mathbb{R}^T \to \mathbb{R}$, parameterized by neural network weights $\theta$, that estimates the *AHI*, a continuous scalar denoted by $\hat{y} \in \mathbb{R}$:

$$\hat{y} = f_\theta(X) \tag{1}$$

The model is trained using a supervised regression objective to minimize the Mean Absoluete Error (MAE) between the

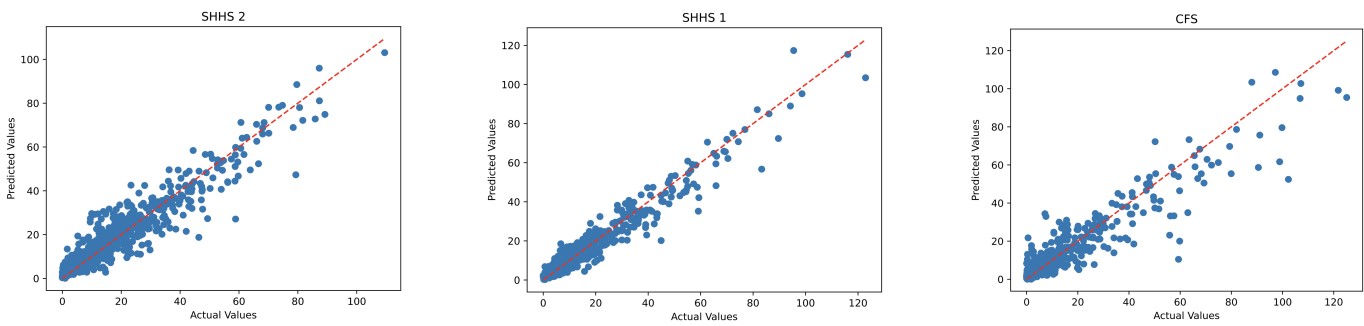

Fig. 2. Scatter plots of predicted vs. ground truth AHI values across three datasets. The dashed red line represents the ideal predictions.

predicted AHI $\hat{y}$ and the ground truth AHI $y$:

$$\mathcal{L}_{\text{reg}} = \frac{1}{N} \sum_{i=1}^{N} |y_i - \hat{y}_i| \tag{2}$$

where $N$ is the number of training samples. Subsequently, the predicted AHI are discretized into standard clinical OSA severity classes using clinically predefined thresholds.

### B. Network Architecture

The proposed architecture is constructed to support both high predictive performance and interpretability by incorporating hierarchical processing modules tailored to different temporal and contextual aspects of the $SpO_2$ signal. The model integrates convolutional layers for local feature extraction, a Squeeze-and-Excitation (SE) block for adaptive channel weighting, bidirectional LSTM layers for capturing sequential dependencies, GATs for modeling relational structure across time, and a MHA mechanism for global interpretability.

#### 1) Feature Extraction

The model takes as input the normalized $SpO_2$ time series $x_{oxi}$ and processes it through four consecutive convolutional blocks designed to extract short-term temporal features that often correspond to desaturation patterns linked to apneic events. Each block contains a 1D convolutional layer with 256 filters and a kernel size of 6, applied to preserve temporal resolution. This is followed by batch normalization to stabilize and accelerate the learning process, and a LeakyReLU activation function with a negative slope coefficient of 0.15 to maintain gradient flow through non-active regions. Temporal downsampling is then performed using average pooling with a pool size and stride of 2, reducing sequence length while retaining essential patterns.

To enhance channel-wise feature representations, the output of the convolutional blocks is passed through an SE block, yielding $x_{SE}$. This block applies global average pooling across the temporal dimension to summarize feature activations per channel.

#### 2) Context Modeling

To capture sequential dependencies in the desaturation patterns, the enhanced features $x_{SE}$ are passed through two stacked bidirectional LSTM layers, producing contextual embeddings $x_{BiLSTM}$. These layers allow the model to learn from both past and future contexts at each time step, crucial for detecting apnea events that may exhibit temporal patterns spanning several seconds or minutes.

Following the bidirectional LSTM, the sequence is represented as a temporal graph, where each time step becomes a node and is connected to adjacent or nearby time points. This graph is processed using a GAT.

#### 3) Attention-Based Representation

The GAT module comprises three stacked GAT layers, each with 4 attention heads and exponential linear unit activation. These layers produce successive embeddings $\{x_{GAT}^{(1)}, x_{GAT}^{(2)}, x_{GAT}^{(3)}\}$, where each $x_{GAT}^{(i)}$ represents the contextualized node features after the $i$-th GAT transformation. GATs enable the model to dynamically learn which time steps (nodes) are most relevant to each other by assigning attention weights to their edges. This is particularly valuable for $SpO_2$ signals, where apnea events have nonuniform durations and spacing, and contextual dependencies may span noncontiguous segments. In this implementation, each time step is treated as a node with learned embeddings propagated through the attention heads. The attention mechanism assigns edge weights based on content similarity, capturing higher-level temporal structures such as clustered desaturation events or irregular rhythms. By aggregating neighboring features through these learned weights, the model adaptively highlights time regions critical for OSA severity. To further enhance learning, the intermediate outputs $\{x_{GAT}^{(1)}, x_{GAT}^{(2)}, x_{GAT}^{(3)}\}$ are concatenated with the BiLSTM representation $x_{BiLSTM}$, forming a residual connection that preserves information across stages and improves gradient flow. The resulting fused representation is denoted as $x_{concat}$

To complement the localized focus of the GAT, the concatenated features $x_{concat}$ is passed through an MHA layer for global sequence modeling and interpretability. This module contains four parallel attention heads, each with a key/query dimension of 16. Unlike GAT, which focuses on near-neighbor relationships, MHA is capable of capturing long-range dependencies across the entire sequence. Each head in MHA learns to focus on different patterns in the signal, such as periodic dips, slow recovery segments, or compound desaturations. By operating in parallel, these heads offer a more comprehensive view of the input's temporal structure. The output of this module, denoted as $x_{MHA}$, is then summarized via an attention pooling layer, which computes a weighted sum of feature vectors across time. This step produces a compact sequence-level representation $x_{pool}$ that emphasizes the most informative time points for AHI prediction.

In parallel, a residual block processes the raw $SpO_2$ input $x_{oxi}$ to preserve coarse-grained signal trends that might be diminished in the deeper layers. This block applies global average pooling, followed by a dense layer with batch normalization and LeakyReLU activation, generating a low-dimensional summary $x_{res}$. The pooled attention representation $x_{pool}$ and the residual output $x_{res}$ are concatenated to form the fused

TABLE I
SUMMARY OF CLASSIFICATION ACROSS VARIOUS TEST DATASETS. VALUES
INCLUDE 95% CONFIDENCE INTERVALS.

| Metric | SHHS-1 | SHHS-2 | CFS |
|---|---|---|---|
| F1 Score | $0.810 \pm 0.032$ | $0.741 \pm 0.030$ | $0.791 \pm 0.030$ |
| Precision | $0.814 \pm 0.031$ | $0.750 \pm 0.029$ | $0.810 \pm 0.029$ |
| Recall | $0.809 \pm 0.035$ | $0.741 \pm 0.031$ | $0.783 \pm 0.031$ |
| Sensitivity | $0.809 \pm 0.032$ | $0.741 \pm 0.031$ | $0.783 \pm 0.032$ |
| Specificity | $0.926 \pm 0.014$ | $0.898 \pm 0.014$ | $0.922 \pm 0.018$ |
| $R^2$ | 0.941 | 0.890 | 0.868 |
| ICC | 0.9689 | 0.9308 | 0.9237 |

representation $x_{fusion}$. Finally, $x_{fusion}$ is passed through a fully connected layer to generate the final latent representation, from which a single scalar output—the predicted AHI $\hat{y}$ is obtained. This fusion of graph-based, attention-based, and residual features ensures that the model leverages both fine-grained desaturation patterns and broader physiological trends, while maintaining interpretability through its attention mechanisms.

## IV. EXPERIMENTAL RESULTS AND DISCUSSION

### A. Datasets, Preprocessing and Computational Resources

I. SHHS Dataset: The Sleep Heart Health Study (SHHS) is a multi-center cohort study funded by the National Heart, Lung, and Blood Institute (NHLBI) to investigate the health impacts of sleep-disordered breathing, including its links to cardiovascular disease and mortality [20], [21]. SHHS-1 includes baseline data from 6,441 participants (1995–1998), with polysomnography (PSG) data available for 5,793 individuals. Only the oximetry channel (SaO2), sampled at 1 Hz using a Nonin XPOD 3011 sensor, was used for model training. SHHS-2, collected between 2001 and 2003 from 3,295 participants, served as a validation dataset.

II. CFS Dataset: The Cleveland Family Study (CFS) is a longitudinal study of sleep apnea in 2,284 individuals from 361 families [20], [22]. Data from Visit 5, which includes full overnight PSG with 14 channels (e.g., EEG, airflow, oxygen saturation), was used as an additional validation set to evaluate generalizability.

Before preprocessing, we used a 65/25/10 split for training, validation, and testing, respectively, for the SHHS-1 dataset. Datasets SHHS-2 and CFS were used exclusively as out-of-distribution test datasets. The SHHS-1 testing subset is an in-distribution test dataset. To ensure signal consistency and improve data quality for model training, a comprehensive preprocessing pipeline was applied to the $SpO_2$ signals. All signals were standardized to a fixed length of 25,200 data points by zero-padding or truncation. High-frequency noise was reduced using the Savitzky-Golay filter, which smooths signals while preserving underlying trends. Missing or non-physiological values were imputed using linear interpolation. Signals were then normalized to zero mean and unit variance to eliminate scale differences. Unlike conventional methods that use windowed segments, the entire signal was used as a single sequence to retain long-term temporal dependencies and clinically relevant patterns.

Model training and evaluation were primarily performed on the SHHS-1 dataset using TensorFlow 2.15.0, NVIDIA H200

GPU, 10-core CPU, and 128 GB of RAM. The complete training process took approximately 50 minutes.

### B. Regression & Classification

To quantify the performance of our proposed model, we evaluated both the estimation of AHI and OSA severity classification across the three test sets. For regression, we evaluate the performance using the coefficient of determination ($R^2$), and for classification, we evaluate the performance using F1 score, precision, recall, sensitivity, and specificity. As shown in Table I, the model demonstrates consistent performance across datasets, achieving high $R^2$ values and strong classification metrics using only single-channel $SpO_2$ signals.

To further gain insight into the performance of our model in estimating the AHI, we plot the scatter plots, and Bland–Altman plots for each dataset, as shown in Figures 2–3. Figure 2 presents scatter plots comparing predicted to the ground truth AHI values. Observation shows that the predictions align closely with the identity line (red dashed line), particularly for SHHS-1, where higher data density improves regression fidelity. SHHS-2 and CFS show greater variance at higher AHI values, likely due to fewer samples in those ranges, yet still maintain a strong linear relationship overall. Lastly, to assess systematic bias and limits of agreement, Figure 3 shows Bland–Altman plots for the three test set. Most of the residuals lie within the 95% confidence interval bounds, confirming consistent model performance across AHI ranges. Although some outliers are observed, especially at extreme AHI values, the mean bias remains close to zero across all datasets, indicating no significant over- or underestimation trend.

Additionally, we assess the model's classification performance across all OSA severity levels using confusion matrices on the test datasets, as shown in Figure 4. These matrices evaluate the model's ability to distinguish between clinically defined OSA severity categories: No Apnea, Mild, Moderate, and Severe. Each matrix presents the percentage of correctly and incorrectly classified samples per class. Strong diagonal dominance across all matrices indicates that the model consistently predicted the correct severity category for the majority of test samples, suggesting its effectiveness in capturing class-specific desaturation patterns from $SpO_2$ signals. Misclassifications were primarily observed between adjacent severity categories, likely due to the proximity of AHI values to class boundaries. This indicates that the model may be sensitive to subtle variations in desaturation severity but can struggle with borderline cases where AHI values fall near diagnostic thresholds. Notably, the model demonstrated high precision and recall for both the Severe and No Apnea classes across all datasets. No Severe cases were misclassified as No Apnea, and vice versa. This may be attributed to the distinctive and frequent desaturation events typical of Severe OSA cases, which are absent in No Apnea subjects. In contrast, the Mild and Moderate classes exhibited more mutual misclassification, likely due to less distinctive signal features and physiological overlap. These results support the model's reliability in distinguishing clinically significant cases while also highlighting areas where diagnostic ambiguity is inherent to the data.

### C. Comparison to Prior Work

Figure 5 presents a comparison of key evaluation metrics, including sensitivity, precision, F1 score, and $R^2$ against existing $SpO_2$-based models reported in the literature [9], [23]. The

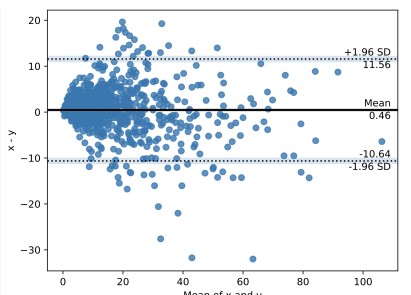 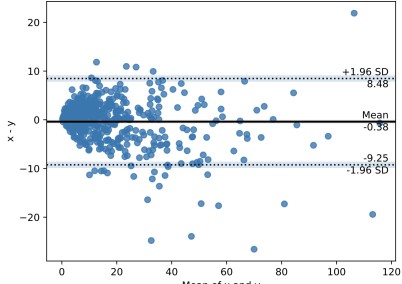 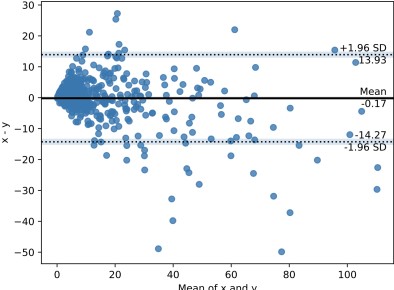

Fig. 3. Bland-Altman plots for the three test sets (SHHS-2 plot on the left, SHHS-1 plot on the center, and CFS plot on the right), showing the difference between predicted and true AHI against their mean. Majority of the residuals lie within the confidence interval with some outliers.

error bars indicate the standard deviation, capturing variability in performance. Levy et al. developed a multimodal deep learning model called OxiNet to estimate AHI using patients' clinical data and single-channel oximetry signals [9], while Chen et al. [23] developed a single-channel oximetry-based deep neural network (DNN) model for AHI estimation. OxiNet and DNN were used as baseline models. Both our approach and the baseline models used the complete oximetry signal without partitioning it into smaller windows. To ensure a fair and consistent evaluation, we intentionally selected baseline methods that also rely on full-sequence processing. We observed that our model consistently outperformed these baselines, achieving the highest $R^2$ and marginally better values for all classification metrics. These findings indicate strong generalization to unseen data while preserving a balanced F1 score. In several cases, our model shows slightly lower standard deviation bars, suggesting somewhat more consistent predictions. The 1 Hz single-channel input and compact architecture make the model well-suited for deployment in resource-constrained environments. These results underscore the robustness and reliability of our approach for OSA severity classification using single-channel $SpO_2$.

### D. Ablation Study

To assess the significance of each model component, an ablation study was conducted to evaluate the contributions of the attention mechanisms, specifically GAT and MHA. The performance of the proposed model was compared against variants of the model with GAT removed, MHA removed, both components removed, and residual connections removed. This was carried out using the SHHS-1 dataset. The results of the ablation study, summarized in Table II, provide insights into the importance of GAT and MHA.

Removing GAT resulted in a noticeable decline across all performance metrics. As GAT is capable of learning complex interdependencies, its absence hinders the model's ability to capture important temporal relationships. Similarly, removing the MHA layer also led to a marked decrease in performance, underscoring its essential role in accurate AHI prediction. Without MHA, the model appears less effective at attending to the most relevant oximetry patterns associated with OSA. The greatest performance degradation was observed when both GAT and MHA were removed. GAT captures intricate signal relationships, while MHA enhances this representation by focusing on the most informative segments. Furthermore, to assess the contribution of the residual connections, we performed an additional ablation by removing them from the architecture while keeping MHA and GAT. The resulting

performance dropped across all metrics compared to the full proposed model, but it remained higher than the performance without MHA and GAT. Fisher's Z-transformation test yielded a p-value of 0.0047, indicating that the performance drop observed when both MHA and GAT were removed is statistically significant compared to the full proposed model. This suggests that while the residual connections enhance model stability and performance, the primary gains are from the attention mechanisms. Together, these components form a robust pipeline that outperforms models relying solely on conventional convolutional and recurrent layers.

TABLE II
RESULTS OF ABLATION STUDY HIGHLIGHTING THE INDIVIDUAL
CONTRIBUTIONS OF EACH KEY COMPONENT TO THE OVERALL MODEL
PERFORMANCE.

| Ablation | F1 Score | Precision | Recall | Sensitivity | Specificity | $R^2$ | ICC |
|---|---|---|---|---|---|---|---|
| Without GAT+MHA | $0.754 \pm 0.033$ | $0.762 \pm 0.033$ | $0.753 \pm 0.033$ | $0.754 \pm 0.037$ | $0.903 \pm 0.016$ | 0.916 | 0.9567 |
| Without GAT | $0.788 \pm 0.032$ | $0.801 \pm 0.030$ | $0.785 \pm 0.032$ | $0.785 \pm 0.034$ | $0.921 \pm 0.013$ | 0.922 | 0.9613 |
| Without MHA | $0.778 \pm 0.037$ | $0.786 \pm 0.032$ | $0.780 \pm 0.035$ | $0.781 \pm 0.035$ | $0.914 \pm 0.014$ | 0.929 | 0.9678 |
| Without Residual Layer | $0.765 \pm 0.033$ | $0.774 \pm 0.033$ | $0.768 \pm 0.031$ | $0.769 \pm 0.034$ | $0.907 \pm 0.015$ | 0.917 | 0.9642 |
| Proposed Model | $0.810 \pm 0.032$ | $0.814 \pm 0.031$ | $0.809 \pm 0.035$ | $0.809 \pm 0.032$ | $0.926 \pm 0.014$ | 0.941 | 0.9689 |

### E. Attention-Based Explainability & Discussion

To assess the interpretability of the model, we visualized the average attention matrices generated by the MHA layer, grouped by OSA severity class, as shown in Figure 6. Each matrix represents the averaged attention scores across all test samples within a given class. These scores reflect how much the model attends to each key time step when processing a corresponding query time step. Diagonal elements indicate self-attention, while off-diagonal elements capture interactions between different time points in the oximetry signal. We observed from the Mild class that the matrix exhibits strong diagonal dominance with a few scattered vertical lines. A pronounced diagonal suggests that the model primarily attends to each time step individually, consistent with the relatively stable $SpO_2$ signal characteristic of mild OSA, where apneic events are infrequent and subtle. Additionally, in the Moderate class, the diagonal dominance persists but is accompanied by more frequent and distinct vertical clusters scattered across the matrix. This pattern indicates that the model is attending both to individual time steps and to specific regions of the sequence. This is consistent with moderate OSA, which involves more significant but still intermittent desaturation events spaced across the night. The Severe class matrix displays numerous bright clusters throughout, indicating that the model is linking multiple segments of the signal. These bright regions suggest frequent and repetitive desaturation episodes. Additionally, the presence of alternating dark and bright clusters may reflect

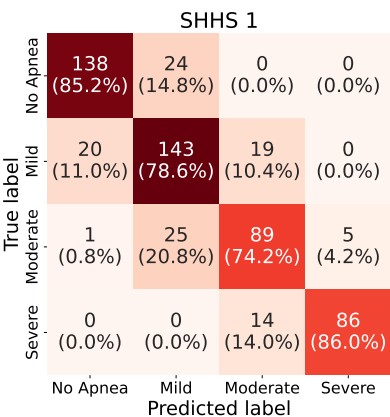 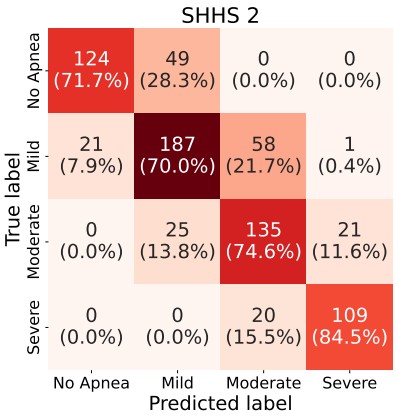 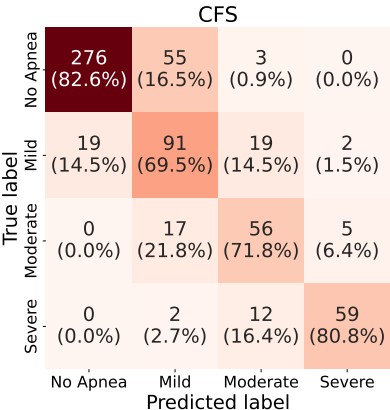

Fig. 4. The confusion matrix results of our framework were evaluated using three different testing datasets (SHHS 1, SHHS 2, and CFS) to assess generalization. The results demonstrate varying classification outcomes across the four predicted classes.

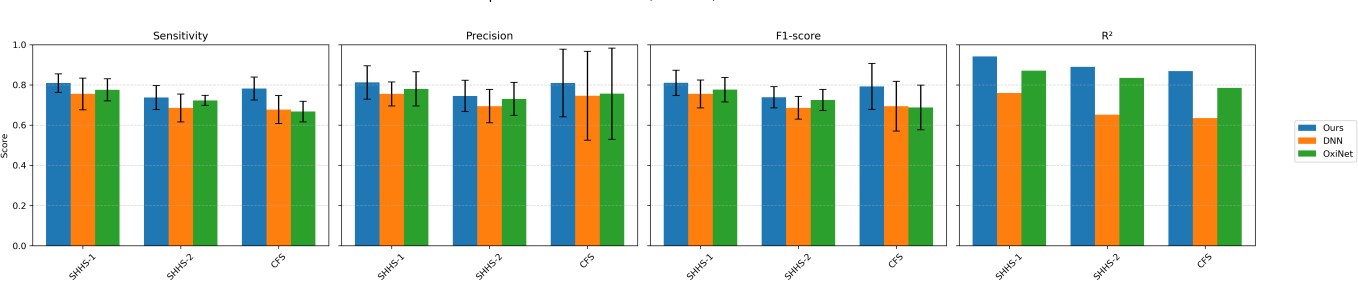

Fig. 5. Comparison of Sensitivity, Precision, $R^2$, and F1 Across Models and Datasets with standard deviation. Our model consistently outperformed the other models across all three test sets.

recurring desaturation-resaturation cycles, which are typical in severe OSA cases.

Interestingly, the matrix for the No Apnea class reveals elevated attention toward the end of the sequence, with minimal activity elsewhere. A strong diagonal with few or no other clusters aligns with expectations for normal breathing, where the signal lacks disruptive events. However, the localized attention toward the sequence's end could correspond to brief desaturation-recovery cycles that occur during REM sleep, even in healthy individuals [24]. This observation aligns with prior studies reporting that mild desaturations are more frequent during early morning REM stages [25].

To better understand how attention influences the estimation of Apnea versus No Apnea, Figure 7 presents a side-by-side comparison of the average attention matrices generated by MHA for the No Apnea and Apnea classes, with a cutoff threshold set at AHI value of 5. The attention values ranging between 0.000575-0.000775 are raw, unnormalized outputs from the MHA module. The Apnea class displays multiple bright vertical clusters distributed throughout the sequence, indicating that the model attends to several distinct time segments. These regions likely correspond to repeated apnea events, aligning with the clinical profile of sleep apnea. The clear contrast between the No Apnea and Apnea patterns demonstrates the model's ability to effectively distinguish between these two conditions.

These attention patterns support the model's interpretability and its ability to adaptively focus on class-specific temporal regions within the oximetry signal. The combination of con-

volutional layers, bidirectional LSTM, GAT, and MHA allows the model to capture both local and global dependencies, while residual connections help preserve underlying signal trends.

Nonetheless, this study has certain limitations. While attention matrices provide qualitative insight into the model's focus, analyzing class-specific attention distributions or further validation against clinically annotated event timings is necessary to confirm their alignment with expert-defined events. Our future work will also explore the influence of borderline AHI values on misclassification rates, particularly for adjacent severity classes, by using an uncertainty-aware framework to improve the model's reliability in ambiguous cases.

## V. CONCLUSION

In this study, we present an interpretable deep learning framework for estimating the AHI and classifying OSA severity using single-channel $SpO_2$. Our pipeline integrates convolutional layers, bidirectional LSTMs, GAT, and MHA to effectively capture both local and global temporal dependencies in the $SpO_2$ signal. Extensive evaluation across two large, diverse sleep study datasets demonstrates our model's strong predictive performance, generalizability, and robustness across OSA severity levels. Beyond predictive accuracy, a key contribution of this work is model interpretability. The use of attention mechanisms allows for intuitive visualization of the model's focus, revealing class-specific temporal patterns that align with clinical expectations. These insights enhance the model's transparency and support its potential integration into clinical decision-making. Overall, this work advances the development of accurate and explainable AI tools for sleep apnea screening

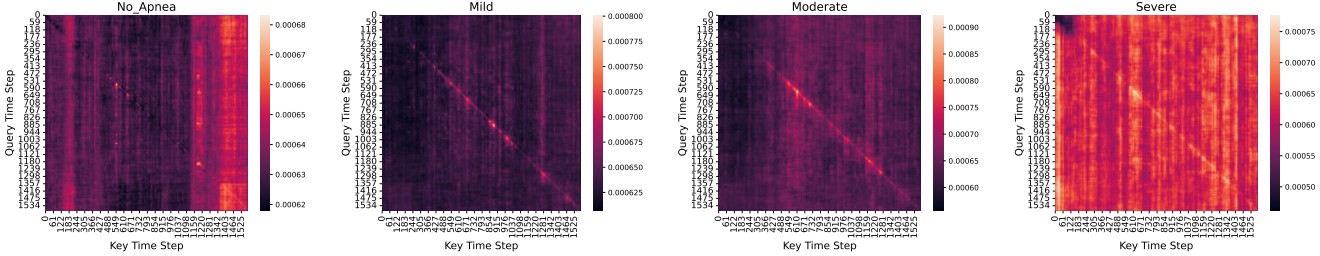

Fig. 6. Average attention matrices for each AHI severity class. Brighter values indicate higher attention weights between time steps. Patterns show class-dependent differences in focus.

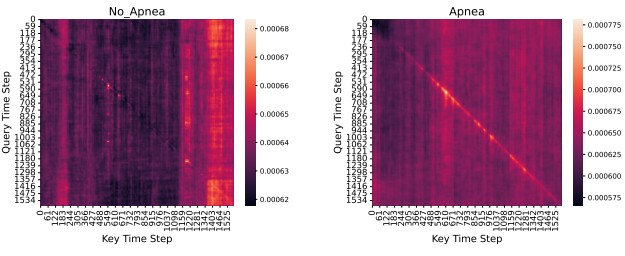

Fig. 7. Average attention matrices for No Apnea and Apnea classes (AHI cutoff set at 5). Brighter values indicate higher attention weights between time steps. Patterns show class-dependent differences in focus.

using accessible, non-invasive physiological data, and lays the foundation for broader adoption of interpretable models in digital health.

## ACKNOWLEDGMENT

This research was supported by a seed research grant from Shriners Children's Hospital. Additional support was provided in part by the AI Makerspace of the College of Engineering and other research cyberinfrastructure resources and services offered by the Partnership for an Advanced Computing Environment (PACE) at the Georgia Institute of Technology, Atlanta, Georgia, USA. We also gratefully acknowledge Wallace H. Coulter Distinguished Faculty Fellowship, a Petit Institute Faculty Fellowship, and research funding from Amazon and Microsoft Research to Professor May D. Wang.

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
