# OpenReview forum: "Developing An Attention-Based Deep Learning Framework for Obstructive Sleep Apnea Detection Using Single-Channel Oximetry Signal"
_IEEE.org/EMBS/BHI/2025/Conference — BHI 2025_

### Official Review · Reviewer_65pe · 2025-07-16
**This paper proposes an interpretable (through attention-based visualizations) deep learning framework for obstructive sleep apnea (OSA) detection and severity classification using only a single-channel pulse oximetry (SpO2) signal. The model integrates convolutional layers, bidirectional LSTM, Graph Attention Networks (GAT), and Multi-Head Attention (MHA) to capture temporal patterns. The model was evaluated on three large datasets (SHHS-1, SHHS-2, and CFS).**

**Confidence:** 5
**Clarity Of Writing:** great
**Clinical Significance:** great
**Methodological Novelty:** great
**Overall Rating:** 7

**Experiments And Results:**

great

**Questions For The Authors:**

- Page 3: "... to minimize the Mean Absoluete Error (MAE) between the predicted AHI ...": But the Eq. 2 is MSE not MAE as mentioned in the text.
- Page 4: The preprocessing steps were applied to the entire dataset (both training and testing splits) prior to splitting, which can lead to overestimated final performance metrics. This approach may introduce data leakage, ultimately compromising the reliability of the results.
- Page 5: "We observed that our model consistently outperformed these baselines, achieving the highest R2 and marginally better values for all classification metrics.": Are the differences statistically significant?
- Page 5: "These findings indicate strong generalization to unseen data while ...": But the test data has been seen already in the preprocessing step (data leakage).
- Page 5: "Removing GAT resulted in a noticeable decline across all performance metrics.": Is it statistically significant?
- Table II: Please bold the best values for each performance metric.
- Fig. 5: To facilitate more informative comparisons and visualizations, please include standard deviation values in the bar diagrams.
- Page 6: "... with a cutoff threshold set at 5.": What threshold values were used, given that the attention matrix values are relatively low (0.0006) (i.e., it is expected to see a cutoff threshold set around 0.0006, not 5)?

**Strengths:**

- The paper addresses a significant public health issue (underdiagnosed OSA & costly detection using PSG) by offering a non-invasive, low-cost, and scalable screening method using solely a single-channel SpO2.
- The proposed deep learning framework uniquely combines convolution, bidirectional LSTM, Graph Attention Networks (GAT), and Multi-Head Attention (MHA) modules.
- The focus on model interpretability (providing meaningful insights into desaturation patterns across different OSA severity levels) through attention-based visualizations.
- The paper provides a thorough comparison with existing SpO2-based models (e.g., OxiNet & Chen's DNN model).
- The ablation study systematically quantifies the contribution of GAT and MHA modules.

**Summary Of The Paper:**

The paper addresses the underdiagnosis of Obstructive Sleep Apnea (OSA) by proposing a deep learning model that utilizes single-channel pulse oximetry (SpO2) signals. The proposed model aims to estimate the Apnea-Hypopnea Index (AHI) and classify OSA severity. The model includes convolutional (Conv1D) layers and a Squeeze-and-Excitation (SE) block to extract short-term temporal features, bidirectional LSTM layers to capture sequential dependencies and learn from both past and future contexts, Graph Attention Network (GAT) to learn dynamic attention weights between time steps, capturing non-uniform duration and spacing of apnea events, Residual connections to preserve information from previous layers and to retain coarse-grained signal trends, and a Multi-Head Attention (MHA) layer for global sequence modeling and interpretability, focusing on different patterns and long-range dependencies. The model is trained to minimize the Mean Absolute Error (MAE). The interpretability of the model is assessed by visualizing attention matrices, which show how the model focuses on different time steps for various OSA severity classes. The model was evaluated on the SHHS-1, SHHS-2, and CFS datasets employing regression (R2) and classification metrics (F1 score, precision, recall, sensitivity, and specificity). The authors highlight the model's ability to distinguish between No Apnea and Severe cases (extreme categories) particularly well, while noting some misclassifications between adjacent mild and moderate categories. An ablation study confirmed the importance of GAT and MHA components for the model's performance.

**Weaknesses:**

- The interpretation of attention heatmaps (Figures 6 and 7) is largely qualitative, relying on visual inspection, while it would benefit from more quantitative analysis.
- The misclassifications between adjacent severity categories (particularly Mild and Moderate).
- The code reproducing the results and figures is not publicly available.

---

### Official Review · Reviewer_cHwS · 2025-07-16
**This study presents a robust and interpretable deep learning framework for OSA (Obstructive Sleep Apnea) severity classification and AHI estimation using only single-channel SpO₂ data**

**Confidence:** 4
**Clarity Of Writing:** good
**Clinical Significance:** good
**Methodological Novelty:** good
**Overall Rating:** 7

**Experiments And Results:**

good

**Questions For The Authors:**

Good Work!

**Strengths:**

This study introduces a novel and interpretable deep learning framework that accurately estimates AHI and classifies OSA severity using only single-channel SpO₂ data, leveraging convolutional, recurrent, and attention-based architectures. Its strong performance across multiple datasets, combined with meaningful interpretability through attention visualizations, highlights its clinical relevance and potential for scalable, non-invasive sleep apnea screening

Provided experiments with two large datasets.

Good visualization of the results

**Summary Of The Paper:**

This study presents a robust and interpretable deep learning framework for OSA (Obstructive Sleep Apnea) severity classification and AHI estimation using only single-channel SpO₂ data, demonstrating strong performance across multiple large-scale datasets. The integration of advanced attention mechanisms and interpretability tools enhances both the accuracy and clinical applicability of the proposed model for scalable OSA screening.

**Weaknesses:**

While the study demonstrates promising results, the interpretability offered by attention matrices would benefit from further validation against clinically annotated apnea event timings. Future work addressing misclassification around borderline AHI values, especially between adjacent severity classes, will enhance model reliability in real-world clinical scenarios

---

### Official Review · Reviewer_NXWW · 2025-07-17
**Interesting topic and technically sound architecture, while explainability remains limited**

**Confidence:** 4
**Clarity Of Writing:** good
**Clinical Significance:** great
**Methodological Novelty:** good
**Overall Rating:** 5

**Experiments And Results:**

good

**Questions For The Authors:**

All points listed in the weaknesses section.

**Strengths:**

- Leveraging oximetry signals to detect sleep apnea is an interesting, important, and practical approach with clear real-world implications.

- The proposed framework is technically sound, with a reasonable architecture.

- The reported performance appears promising and demonstrates potential for real-world application.

**Summary Of The Paper:**

This paper proposed a deep learning based framework for sleep apnea detection leveraging SPO2 signals.

**Weaknesses:**

- The framework includes several complex components (e.g., BiLSTM and residual layers), but the computational cost is not discussed. It is unclear whether the model is lightweight enough for deployment on resource-constrained wearable devices.

- The comparison with existing works lacks clarity. For instance, it is not specified whether references [8] and [21] use the same dataset and evaluation metrics, which is crucial for fair benchmarking.

- The explainability of the model remains limited. Although attention maps are included, it is still unclear how the parameter-heavy architecture interprets the SpO₂ signal and which features contribute most to the final prediction.

- The font size in Figures 1, 6, and 7 is too small, making them difficult to read. Enhancing the figure clarity would improve the presentation quality.

---

### Official Review · Reviewer_9j61 · 2025-07-21
**an interpretable, attention-based deep learning framework for estimating Apnea-Hypopnea Index (AHI) and classifying Obstructive Sleep Apnea (OSA) severity using single-channel pulse oximetry (SpO₂) signals.**

**Confidence:** 4
**Clarity Of Writing:** great
**Clinical Significance:** great
**Methodological Novelty:** good
**Overall Rating:** 4
**Final Rating:** 5

**Experiments And Results:**

good

**Questions For The Authors:**

- Have you considered including an ablation of the residual layer? Given its likely contribution to performance and stability, including this comparison would provide a more complete picture of the architectural contributions.
- Could you clarify the exact novelty in applying GAT and MHA to this task?
- What is the rationale for choosing full-length SpO₂ signals vs. windowed inputs? Did you try time-windowed models (e.g., 30-second epochs)? Full-sequence processing is computationally expensive and may limit generalization. A sensitivity analysis on this design choice would be insightful.

**Strengths:**

- Well-Motivated Clinical Impact: The focus on single-channel SpO₂ provides a low-cost, scalable alternative to full PSG, addressing a critical barrier to mass OSA screening.
- Interpretability: Visualization of attention matrices across OSA severity levels provides meaningful clinical insights and aligns with recent demands for explainable AI in healthcare.
-

**Summary Of The Paper:**

The submitted manuscript proposes an interpretable, attention-based deep learning framework for estimating Apnea-Hypopnea Index (AHI) and classifying Obstructive Sleep Apnea (OSA) severity using single-channel pulse oximetry (SpO₂) signals. The architecture combines Conv1D, bidirectional LSTM, Graph Attention Networks (GAT), and Multi-Head Attention (MHA) to capture both local and global signal patterns. The model is evaluated across three large datasets (SHHS-1, SHHS-2, and CFS), achieving high R² and classification performance. It further emphasizes interpretability via attention visualization. An ablation study is presented to evaluate the contribution of GAT and MHA components.

**Weaknesses:**

- Marginal Gain Interpretation: While the model outperforms OxiNet and others, the performance gains (especially for classification metrics) are modest. It is unclear whether improvements stem from architectural innovations (GAT/MHA) or simply a more complex model with more parameters. This weakens the novelty claim.
- Unclear Novelty of GAT Use: GAT has previously been applied in biomedical signal modeling. The paper would benefit from clarifying why its application here is significantly novel—especially given the narrow scope (single-channel SpO₂).
- Incomplete Ablation Study – Missing Residual Layer Impact:
While the authors perform an ablation study removing GAT and MHA components, they do not evaluate the effect of the residual connections. Given that residual layers are known to significantly impact gradient flow and performance in deep architectures, it is important to quantify their contribution. Without this, it is unclear whether the model’s improvements stem from the attention mechanisms themselves or from architectural depth stabilization via residual learning.